# Can Biofertilizers Reduce Synthetic Fertilizer Application Rates in Cereal Production in Mexico?



Jesús Santillano-Cázares [1], Marie-Soleil Turmel [2,3], María Elena Cárdenas-Castañeda [2], Santiago Mendoza-Pérez [4], Agustín Limón-Ortega [5], Roberto Paredes-Melesio [6], Luis Guerra-Zitlalapa [2] and Iván Ortiz-Monasterio [2,*]

1   Instituto de Ciencias Agrícolas, Universidad Autónoma de Baja California, Carretera a Delta S/N, Ejido Nuevo León, Mexicali 21705, Mexico; jsantillano@uabc.edu.mx
2   Centro Internacional de Mejoramiento de Maíz y Trigo (CIMMYT), Km. 45, Carretera Mexico-Veracruz, El Batan, Texcoco 56130, Mexico; marie-soleil.turmel@crs.org (M.-S.T.); m.cardenas@cgiar.org (M.E.C.-C.); l.guerra@cgiar.org (L.G.-Z.)
3   Catholic Relief Services, Baltimore, MD 21201, USA
4   Facultad de Ciencias Agronómicas, Universidad Autónoma de Chiapas, Carretera Ocozocoautla-Villaflores Km 84.5, Apartado Postal 78, Villaflores 30470, Mexico; stgo.mepe84@hotmail.com
5   Instituto Nacional de Investigaciones Forestales, Agrícolas y Pecuarias, Km 13.5 Los Reyes−Texcoco, Edo, Texcoco 56250, Mexico; limon.agustin@inifap.gob.mx
6   MUNSA Molinos S. A. de C.V., Edificio JDB, Avenida Circunvalación Agustín Yáñez, No. 2583, Colonia Arcos Vallarta, Guadalajara 44130, Mexico; Paredes.roberto59@gmail.com
*   Correspondence: i.ortiz-monasterio@cgiar.org; Tel.: +52-55-5804-2004

**Abstract:** Biofertilizers are considered as potential supplements or alternatives to fertilizers. The objective of the present study is to evaluate different biofertilizers in combination with synthetic fertilizers on the yields of maize and wheat in several states in Mexico. Fourteen biofertilizer treatments plus a treatment with 100% the locally recommended fertilizer rate (RFR), another with 50% RFR (the control treatment), and one without any fertilizer (for a total of 17 treatments) were tested on maize and wheat in five states across Mexico. Field experiments were established in five states and several years for a total of 14 experiments in Mexico. In general, except for the experiments conducted in moderately low soil P conditions, Chiapas and Sonora (maize), no response to biofertilizers was observed in the remaining locations, through the years in wheat and maize. We conclude that in high input production systems, the biofertilizer response is more an exception than a rule with only 21% of the experiments showing a significant difference in favor of biofertilizers and only 4 of 15 products tested produced a yield response in more nitrogen deficient environments. Some products containing AMF may be beneficial in maize production systems with phosphorus deficient environments.

**Keywords:** corn; wheat; organic fertilizers; Guanajuato; Chiapas; Tlaxcala; Sonora; Campeche

## 1. Introduction

The use of biofertilizers in Mexico dates back to pre-Columbian times, as mud from lakes (located near what today is Mexico City) loaded with a variety of microorganisms was used to build floating plots (called chinampas) to grow crops [1]. More recently, Armenta-Bojorquez et al. [2] indicated that the state of Sinaloa (a highly productive, high input agricultural state in northwestern Mexico) widely adopted the use of biofertilizers for N fixation on legume crops around the 1970s and 1980s. Currently the irrigated intensive production systems in Sinaloa, for the most part, relay on synthetic fertilizers. In contrast, the state of Chiapas, which has fewer input intensive systems, and mostly rainfed agriculture, in southern Mexico, seems to be one of the most enthusiastic and successful states for testing biofertilizers [3,4]. Today, with the goal

of improving productivity and reducing the costs of production while minimizing environmental impact, the Mexican government promotes the use of biofertilizers across the whole country. They see biofertilizers as a way to cut down the use of synthetic fertilizers; regardless of obvious agroecological and input use differences across a highly diverse country (https://www.gob.mx/agricultura%7Cregionlagunera/articulos/sagarpa-entrego-2-9-toneladas-de-biofertilizantes-para-el-mejoramiento-del-suelo (accessed on 24 December 2021)). Thus, given the present wave of interest, farmers in Mexico are looking for alternative management practices that could help them reduce their fertilization costs hence, making their activity more profitable. Among these management options, the optimization of mineral fertilizer rates, timing, and methods of application are of great importance [5,6], as well as fertilizer sources such as organic fertilizers [7,8] and biofertilizers [9,10].

The term biofertilizer was defined by Vessey [11] as "a substance which contains living microorganisms which, when applied to seed, plant surfaces, or soil, colonizes the rhizosphere or the interior of the plant and promotes growth by increasing the supply or availability of primary nutrients to the host plant". Biofertilizers can contribute to soil fertility through N fixation, phosphorus solubilization, and an extended root system through association with vesicular-arbuscular mycorrhizal fungi in the soil (AMF), which can improve soil exploration for increased water and nutrient uptake; principally for P uptake. This in turn can result in a better tolerance to biotic and abiotic stress, protection against pathogens, and a general increase in plant fitness [12,13]. This technology more commonly involves seed inoculation with bacteria of the genera *Azospirillum* spp. [14,15], *Bacillus* spp. [16,17], and *Pseudomonas* spp. [18,19]; as well as mycorrhizas like *Glomus* spp. [20,21]. However, there are other microorganisms also used as biofertilizers. Itelima et al. [22] have made a detailed list of biofertilizers, distinguishing among those used for nitrogen fixation, phosphate solubilization, phosphate mobilizers, biofertilizers for micronutrients, and plant growth promoting Rhizobacteria. Commercial biofertilizers products can have a specific microorganism or a consortium of several of these microorganisms as granular or liquid presentations to be applied (inoculated) to seeds, soil, or plant tissues [23–25].

The use of biofertilizers emerges as a potential sustainable alternative for improving maize and wheat cropping systems [26,27]. A growing interest in Mexico is arising based on the expectation that using biofertilizers may partially substitute the use of synthetic fertilizers [28–32]. It has been suggested that biofertilizers can save up to 50% of synthetic fertilizers [4,33]. Positive effects from different microorganisms have been reported for laboratory and greenhouse experiments [16,17,34–38]. However, the potential of biofertilizers for decreasing synthetic fertilizer rates and or increasing the yield of cereals under field conditions in Mexico has been limited and unclear. The quality of biofertilizer products has been questioned due to several factors.

Herrmann & Lesueur [39] indicated that " . . . many of the products that are currently available worldwide are often of very poor quality, resulting in the loss of confidence from farmers. The formulation of an inoculant is a crucial multistep process that should result in one or several strains of microorganisms included in a suitable carrier, providing a safe environment to protect them from the often harsh conditions during storage and ensuring survival and establishment after introduction into soils. One of the key issues in formulation development and production is the quality control of the products at each stage of the process." Furthermore, Husen et al. [40] found a series of inconsistencies in a study about biofertilizers quality in Indonesia and concluded that there is an urgent need to improve the current quality standard system of biofertilizers. In a study of the effect of biofertilizers on bean (*Vigna* sp.) production in Pakistan, Zahir et al. [41] concluded that "research for the development of inoculum for different advanced genotypes should be continued and more emphasis should be deployed to develop biofertilizers with efficient strains to use them under different climate and soil conditions". A classic paper addressing the inconsistency and variability of biofertilizers in Mexican agriculture was published by Fuentes-Ramírez and Caballero-Mellado [42]. More recently, other studies conducted in Mexico have recog-

nized having inconsistent results with the use of biofertilizers [28–30]. Even more recently, in 2019, Rodriguez-Ramos et al. [43] in a multi-location trial reported a lack of response of biofertilizers on wheat (*Triticum aestivum* L.) and barley (*Hordeum vulgare* L.) in the Mexican state of Guanajuato. In contrast, there is one study conducted in Chiapas, Mexico [3] that reported definitive advantages of applying biofertilizers together with synthetic fertilizers to increase maize yields. The objective of the present study was to evaluate different commercially available biofertilizers in combination with synthetic fertilizer on the yields of maize and wheat in contrasting climatic and input use conditions in Mexico.

## 2. Materials and Methods

### 2.1. Sites Description

Experiments were established in five Mexican states, distributed across Mexico. The experimental sites were highly contrasting in several agroecologic conditions, such as geographic location, climate, and soil characteristics (except for Campeche; where no soil analyses were performed); as well as input use and agronomic practices (Tables 1 and 2). Solar radiation, precipitation, and the minimum and maximum temperatures occurring during the crop cycle of the experiments in all states and years are presented in Tables 3 and 4. A total of 14 experiments, 12 with maize and 2 with wheat were conducted. Two experiments with maize per state were established in Guanajuato, Chiapas, and Tlaxcala, during 2012 and 2013; two experiments with maize in Campeche, during 2018 and 2019; four experiments with maize in Sonora, during 2016, 2017, 2018, and 2019; and two experiments with wheat in Sonora, in 2018 and 2019.

**Table 1.** State, municipality, coordinates, elevation, annual precipitation, mean annual temperature, and input use level of experimental sites in Mexico, from 2012–2013, and from 2016–2019.

| State | Municipality | Coordinates | Elevation (masl) | Annual Precipitation (mm) | Mean Annual Temperature (°C) | Input Use |
|---|---|---|---|---|---|---|
| Guanajuato | Celaya | 20°35′41.24′ N, 100°49′14.49″ W | 1768 | 603 | 25 | high |
| Chiapas | Villaflores | 16°14′ N, 93°17′ W | 540 | 1183 | 24 | low |
| Tlaxcala (2012) | Huamantla | 19°24′13.12″ N, 97°56′16.23″ W | 2652 | 1183 | 24 | low |
| Tlaxcala (2013) | Huamantla | 19°24′0.34″ N, 97°56′55.85″ W | 2567 | 625 | 15 | low |
| Campeche | Hopelchen | 19°48′25.47″ N, 89°48′39.85″ W | 87 | 1000 | 26 | low |
| Sonora | Cajeme | 27°23′2.92″ N, 109°54′51.09″ W | 40 | 385 | 22 | high |

**Table 2.** Physical and chemical soil characteristics of the experimental sites in Mexico, from 2012–2013, and from 2016–2019.

| | Texture | pH | OM | N-Inorg. | P-Bray | K | Ca | Mg | Na | Fe | Zn | Mn | Cu | B | S | CEC § | Al Sat |
|---|---|---|---|---|---|---|---|---|---|---|---|---|---|---|---|---|---|
| | | | % | | | | | | mg kg$^{-1}$ | | | | | | | mmol kg$^{-1}$ | % |
| Guanajuato | Clay | 8.4 | 2.20 | 18.3 | 16.8 | 1160 | 4671 | 632 | 255 | 10 | 0.9 | 9.8 | 0.76 | 0.7 | 41 | 32.7 | - |
| Chiapas | Sandy Clay Loam | 4.8 | 2.27 | 49.3 | 10.1 | 102 | 508 | 98.4 | 17 | 71 | 2.4 | 49 | 0.64 | 0.4 | 28 | 5.06 | 5.93 |
| Tlaxcala (2012) | Sandy Clay Loam | 6.0 | 1.29 | 3.47 | 18.6 | 264 | 751 | 164 | 15 | 39 | 0.8 | 22 | 1.09 | 0.3 | 56 | 6.01 | - |
| Tlaxcala (2013) | Clay Loam | 6.2 | 0.60 | 3.16 | 10.5 | 162 | 784 | 141 | 16.1 | 22.4 | 0.3 | 13.3 | 0.79 | 0.2 | 20 | Low | - |
| Sonora | Clay | 7.0 | 1.15 | 60 † | 8.7 ‡ | 559 | 8356 | 790 | 621 | 3.6 | 1.0 | 7.7 | 1.0 | - | - | na | - |

† = N total; ‡ = Olsen; § = Cation Exchange Capacity; - = Not available.

**Table 3.** Mean minimum and maximum temperatures, precipitation, and solar radiation occurring during the crop cycles in the experiments in Guanajuato, Chiapas, and Tlaxcala, Mexico.

| Date | Minimum Temperature °C | Maximum Temperature °C | Precipitation (mm) | Solar Radiation (W/m²) |
|---|---|---|---|---|
| **Guanajuato** | | | | |
| May 2012 | 11.3 | 31.2 | 1.9 | 318.1 |
| June 2012 | 13.3 | 28.9 | 11.8 | 288.9 |
| July 2012 | 13.5 | 24.5 | 18.3 | 269.5 |
| August 2012 | 12.7 | 24.9 | 14.1 | 255.6 |
| September 2012 | 12.4 | 24.4 | 12.8 | 247.3 |
| October 2012 | 10.3 | 25.2 | 2.4 | 250.9 |
| May 2013 | 12.8 | 30.4 | 6.3 | 313.6 |
| June 2013 | 13.7 | 28.5 | 14.5 | 300.6 |
| July 2013 | 13.1 | 25.6 | 20.6 | 275.1 |
| August 2013 | 12.5 | 24.9 | 12.5 | 267.6 |
| September 2013 | 13.5 | 23.2 | 16 | 214.9 |
| October 2013 | 12.4 | 23.3 | 15.7 | 224.8 |
| **Chiapas** | | | | |
| June 2012 | 17.9 | 28 | 58.9 | 238.2 |
| July 2012 | 16.2 | 30.1 | 37.9 | 246.9 |
| August 2012 | 17.1 | 28.2 | 71.5 | 209.4 |
| September 2012 | 16.7 | 28.5 | 39 | 238 |
| October 2012 | 16.5 | 27.7 | 26.1 | 231.7 |
| June 2013 | 18.1 | 28.5 | 59.9 | 234.6 |
| July 2013 | 16.3 | 29.6 | 39.9 | 251 |
| August 2013 | 16.7 | 29 | 49.9 | 230.6 |
| September 2013 | 17.3 | 26.6 | 69.4 | 185.7 |
| October 2013 | 17.3 | 27 | 46.7 | 196.6 |
| **Tlaxcala** | | | | |
| April 2012 | 6.4 | 23.2 | 2 | 334.3 |
| May 2012 | 8.1 | 24.6 | 2.5 | 324.8 |
| June 2012 | 8.4 | 22.8 | 7.3 | 286.7 |
| July 2012 | 7.6 | 20 | 9.7 | 289.6 |
| August 2012 | 7.7 | 19.4 | 10.4 | 256.5 |
| September 2012 | 7.8 | 19.8 | 4.5 | 264.1 |
| April 2013 | 8.6 | 26.1 | 5.5 | 324.3 |
| May 2013 | 9.4 | 24.8 | 5.3 | 318 |
| June 2013 | 8.6 | 22.5 | 7.9 | 294.7 |
| July 2013 | 6.8 | 20.8 | 7.9 | 297.2 |
| August 2013 | 7.4 | 20.5 | 4.9 | 271.7 |
| September 2013 | 9.2 | 19.1 | 13.7 | 217.5 |

**Table 4.** Summary of management practices in experimental sites in Mexico, from 2012–2013, and from 2016–2019.

| Site | Water Regime † | Seeding Dates | Crop | Variety † | Seeding Density † | Recommended Fertilizer Rate (kg ha⁻¹) (100%) † | | |
|---|---|---|---|---|---|---|---|---|
| | | | | | | N ‡ | $P_2O_5$ ‡ | $K_2O$ ‡ |
| Guanjuato | irrigation | 5 June 2012 and 22 May 2013 | Maize | Ocelote-Asgrow | 80,000 plants ha⁻¹ | 240 | 60 | 50 |
| Chiapas | rainfed | 29 June 2012, and 18 June 2013 | Maize | P4063W-Pioneer | 62,500 plants ha⁻¹ | 160 | 46 | 30 |
| Tlaxcala (2012) | rainfed | 23 April 2012 | Maize | H-40 INIFAP | 60,000 plants ha⁻¹ | 80 | 50 | 0 |
| Tlaxcala (2013) | rainfed | 18 May 2013 | Maize | H-40 INIFAP | 60,000 plants ha⁻¹ | 80 | 50 | 0 |
| Campeche | rainfed | 5 July 2018, and 10 July 2019 | Maize | WP4082 Pioneer-2018, and CLTHW15002-CIMMYT-2019 | 62,500 plants ha⁻¹ | 90 | 66 | 48 |

**Table 4.** *Cont.*

| Site | Water Regime † | Seeding Dates | Crop | Variety † | Seeding Density † | Recommended Fertilizer Rate (kg ha⁻¹) (100%) † | | |
|---|---|---|---|---|---|---|---|---|
| | | | | | | N ‡ | $P_2O_5$ ‡ | $K_2O$ ‡ |
| Sonora | irrigation | 24 November 2016; 1 December 2017; 14 November 2018; and 15 November 2019 | Maize | Caribu-Asgrow | 100,000 plants ha⁻¹ | 250 | 46 | 0 |
| Sonora | irrigation | 13 December 2018 and 20 December 2019 | Wheat | CIRNO C-2008 and Borlaug 100 | 120 kg ha⁻¹ for CIRNO C-2008, and 100 kg ha⁻¹ for Borlaug-100 | 200 | 46 | 0 |

† When only one regime, variety, seeding density, or recommended fertilizer rates are reported for multiple cycles, it means that they were the same through the cycles. ‡ Synthetic fertilizer sources were urea, diammonium phosphate (DAP), and potassium chloride (KCl).

## 2.2. Crops Management

In each location, locally recommended hybrid maize and wheat varieties were seeded. Depending on the treatments, nitrogen, phosphorus, and potassium fertilizers were applied according to local recommendations for each site. All biofertilizers were applied to the seed prior to planting following the directions prescribed by the products' manufacturers. Weeds and diseases were controlled in each site following local recommendations. The experiments in Sonora and Guanajuato were irrigated, while the experiments in Campeche, Tlaxcala, and Chiapas were established under rainfed conditions. A summary of crops management practices in all sites is presented in Table 4.

## 2.3. Treatment Descriptions and Response Variables

Fourteen biofertilizer treatments was the maximum number that was included in a single study. Not all 14 biofertilizer treatments were tested in each state. Treatments 1–14 were inoculated with biofertilizers and received 50% of the locally recommended fertilizer rates (RFR) (Table 5). Treatments 15, 16, and 17 received 50, 100, and 0%, respectively, of the locally RFR, without biofertilizers. Table 5 shows which biofertilizers were tested in which states. For the full or half the RFR, the synthetic fertilizers consisted of urea as a nitrogen source, diammonium phosphate (DAP) as a phosphorus and nitrogen source, and potassium chloride (KCl), as a potassium source.

**Table 5.** List of treatments by groups or individual states and cycles where each treatment was tested (bottom of the Table), from 2012–2013, and from 2016–2019 in Mexico. The last column in the Table shows the function of biofertilizers in plants [44].

| No. | Company | Treatment † | Organism | Function |
|---|---|---|---|---|
| 1 | Plant Health | Mycor Root Saver | *Entrophospora columbiana*, *Glomus* spp. | Nutrient uptake (principally P and micronutrients) |
| 2 | UNAM | Azofer | *Azospirilum brasilense* | N-fixation/ plant growth promoter |
| 3 | Biofabrica | MicorrizaFer | *Glomus* spp. | Nutrient uptake (principally P and micronutrients) |
| 4 | Tecnologia Agricola Sustentable | FerbiliQ | *Azospirilum brasilense* | N-fixation/ plant growth promoter |
| | | | *Glomus intraradices* | Nutrient uptake (principally P and micronutrients) |
| 5 | El Vergel | Tec-Myc 60 | *Glomus* spp., *Acaulospora scorbiculata, Gigaspora margarita* | Nutrient uptake (principally P and micronutrients) |
| | | | *Bacillus subtillis* | Plant growth promoter/P solubilization |
| | | | *Azosprilum brasilense* | N-fixation/ plant growth promoter |

**Table 5.** *Cont.*

| No. | Company | Treatment † | Organism | Function |
|---|---|---|---|---|
| 6 | BIOqualitum | BactoCROP- TS | *Azospirillum* spp. | N-fixation/ plant growth promoter |
| | | | *Bacillus* spp. | Plant growth promoter/ P solubilization |
| 7 | Universidad Puebla | BiofertiBUAP | *Azospirillum* spp. | N-fixation/ plant growth promoter |
| 8 | Promotora Tecnica Industrial | Bioradix | *Azospirilum brasilense* | N-fixation/ plant growth promoter |
| 9 | INIFAP | Bacteriano 2709 | *Pseudomonas* spp. | P solubilziation/ plant growth promoter |
| 10 | INIFAP | BIOfertilizante | *Glomus intraradices* | Nutrient uptake (principally P and micronutrients) |
| 11 | Universidad Puebla | BiofosfoBUAP | *Pseudomonas* spp. | P solubilziation/ plant growth promoter |
| 12 | Promotora Tecnica Industrial | Spectrum Mico | *Glomus* spp. | Nutrient uptake (principally P and micronutrients) |
| 13 | Promotora Tecnica Industrial | Spectrum Mico Bac | *Glomus* spp. | Nutrient uptake (principally P and micronutrients) |
| | | | *Bacillus* spp. | Plant growth promoter/ P solubilization |
| 14 | Biokrone | Glumix | *Glomus geosporum, Glomus fasciculatum, Glomus constrictum, Glomus tortuosum, Glomus intraradices* | Nutrient uptake (principally P and micronutrients) |
| 15 | | Synthetic fertilizer 50% (control) | | |
| 16 | | Synthetic fertilizer 100% | | |
| 17 | | Non-fertilized | | |

| | | **Tested treatments by Group or individual States/cycles** | | | |
|---|---|---|---|---|---|
| Group 1 | Group 2 | Group 3 | Tlaxcala 2012 | Campeche 2018 ‡ | Campeche 2019 § |
| 1-17 | 1, 2 + 3, 4, 7, 11, and 15–17 | 1, 2 + 3, 4, 7, 11, 7 + 11, and 15–17 | 1-6, 8–11, and 15–17 | 1, 3–5, 8, and 11 | 1, 2 + 3, 7 + 11, and 8 |

† Treatments 1-14 received 50% of the locally recommended fertilizer rates plus biofertilizers; Treatment 15 received 50% of the locally recommended fertilizer rate, without biofertilizers; treatment 16 received the full locally recommended fertilizer rate without biofertilizers; and treatment 17 did not receive any fertilizer. Not all treatments were tested in all locations. ‡ In Campeche in cycle 2018 six biofertilizers were tested plus six synthetic fertilizers varying by brands and rates. § In Campeche in cycle 2019 four biofertilizers were tested plus six synthetic fertilizers varying by brands and rates.

Experiments where the same treatment structures were repeated across states and years were classified in Groups. In Group 1, all 14 biofertilizers plus one treatment with 100% the locally RFR, another with 50% RFR (the control treatment), and one without any fertilizer or biofertilizer (for a total of 17 treatments) were tested on maize and include experiments conducted in Guanajuato (two cycles), Chiapas (two cycles), and Tlaxcala (one cycle 2013). Cycle 2012 in Tlaxcala was not included in Group 1 due to its unique treatment structure, where only 10 out of the 14 biofertilizers were tested. All experiments in Group 1 were replicated three times. Group 2 included four maize experiments in Sonora, where some of these 14 biofertilizers or combinations of these were tested (Table 6). All experiments in Group 2 had four replications. Group 3 included two wheat experiments in Sonora, one experiment per year, where six biofertilizers were tested on two wheat varieties. Experiments in Group 3 were also replicated four times.

**Table 6.** Mean monthly minimum and maximum temperatures, precipitation, and solar radiation occurring during the crop cycles in the experiments in Campeche and Sonora, Mexico.

| Date | Minimum Temperature °C | Maximum Temperature °C | Precipitation (mm) | Solar Radiation (W/m$^2$) |
|---|---|---|---|---|
| **Campeche** | | | | |
| July 2018 | 21.6 | 37 | 8.6 | 277.2 |
| August 2018 | 21.8 | 35.9 | 7.5 | 262.4 |
| September 2018 | 22.2 | 34.1 | 16.4 | 251.5 |
| October 2018 | 21.3 | 32.6 | 11.8 | 232 |
| November 2018 | 19.8 | 31.5 | 5.8 | 197.3 |
| July 2019 | 22.4 | 36.5 | 26 | 275.4 |
| August 2019 | 22.6 | 36.5 | 12.2 | 268.9 |
| September 2019 | 22.2 | 34.7 | 17.6 | 241 |
| October 2019 | 22.4 | 31.7 | 22.2 | 226 |
| November 2019 | 19.7 | 29.2 | 8.9 | 191.1 |
| **Sonora** | | | | |
| | 32 | 13.4 | 0.5 | 174.5 |
| December 2016 | 26 | 10.3 | 2.5 | 126.1 |
| January 2017 | 24.9 | 7.9 | 10.9 | 156.4 |
| February 2017 | 26.6 | 9.5 | 14.7 | 197.9 |
| March 2017 | 29.4 | 10.6 | 0.4 | 255.4 |
| April 2017 | 32.8 | 11.3 | 0 | 295.9 |
| May 2017 | 33.8 | 14.2 | 0 | 321.1 |
| November 2017 | 33.2 | 13.8 | 0.7 | 193.9 |
| December 2017 | 27.1 | 10.2 | 3.9 | 158.6 |
| January 2018 | 27.7 | 7.7 | 0.1 | 190.9 |
| February 2018 | 25.7 | 10.5 | 8.5 | 189.2 |
| March 2018 | 29.1 | 9.9 | 0.4 | 251.1 |
| November 2018 | 29.3 | 11.7 | 0.1 | 183.7 |
| December 2018 | 25.6 | 9.3 | 8 | 152.2 |
| January 2019 | 25.4 | 7.4 | 5.9 | 168 |
| February 2019 | 23.8 | 8.7 | 5.7 | 183.1 |
| March 2019 | 28.4 | 9.3 | 6.5 | 234.6 |
| April 2019 | 30.9 | 11.3 | 0.1 | 287 |
| May 2019 | 32.2 | 12.3 | 0.1 | 309.8 |
| December 2019 | 25.9 | 10.1 | 9.5 | 154.9 |

Unlike the rest of the experiments reported in this paper, in Campeche, the non-fertilized treatment, 100% RFR, or the control (50% RFR + biofertilizers) were not included in the experiments. Instead, five synthetic fertilizers, no biofertilizer containing treatments and seven biofertilizer containing treatments plus a synthetic fertilizer rate of 90-66-48 (N-P-K) were tested in 2018. In contrast, five synthetic and five biofertilizer containing treatments were tested in 2019. Therefore, no grouping was conformed for this state due to the treatment design asymmetries between years. In both 2018 and 2019, the experiments were replicated three times.

Only biofertilizers that were sold commercially and were associated with crop nutrition were included in this study. All biofertilizers had been registered with the Federal Commission for the Protection against Sanitary Risk (COFEPRIS, by its Spanish acronym); an office which is part of the Mexican Federal Government. Crop yields were measured in all sites. Maize yields were adjusted to 14% moisture, while wheat yields were adjusted to 12%.

### 2.4. Experimental Design and Statistical Analyses

Except for the two experiments with wheat in Sonora, which were arranged in a split-plot in a randomized complete block design, the experimental design for all other experiments was a complete randomized block design with three or four replications. All the experiments in Sonora had four replications, for both wheat and maize. All other experiments had three replications. The experiment with wheat in Sonora tested biofertilizers on

two wheat varieties. Maize plot size consisted of eight rows separated by 0.75 m (6 m wide) by 5 m long. Wheat plot size consisted of four beds, 80 cm apart with three rows of wheat on each bed and 5 m long.

Statistical analyses were performed by groups of experiments that shared the same treatment structure or individually when experiments had a unique treatment structure. Analyses of variance (ANOVA's) were performed using PROC GLM; in the statistical package SAS, version 9.2 for Windows (SAS Institute, 2008). Treatment mean separations were performed using the Fisher's least significant difference (LSD) test at a *p* level of 0.05. In Campeche, given its treatment structure, which lacked a control treatment, such as 50% RFR, in addition to means separation, the effectiveness of biofertilizers (plus 50% RFR) was done through a single degree of freedom contrast between all the treatments with biofertilizers plus 90-66-48 (N-P-K) synthetic fertilizer versus treatment number 6 (in 2018) and number 5 (in 2019) (pure synthetic fertilizer at a rate of 90-66-48, N-P-K), one contrast for each cycle.

## 3. Results

*ANOVA's and Mean Comparisons*

ANOVAs for all grouped and non-grouped sites are shown in Table 7. The control treatment to test the effectiveness of biofertilizers was the 50% of local RFR, thus, the core comparison throughout this paper is the 50% RFR vs. the 50% RFR plus the different biofertilizers. From now on, the biofertilizer containing treatments are referred simply by their commercial names.

**Table 7.** ANOVA's of experiments conducted in five Mexican states to test the effectiveness of partially substituting synthetic fertilizers by biofertilizers.

| Grouped Experiments | | | | | | | | | | | |
|---|---|---|---|---|---|---|---|---|---|---|---|
| **Group 1: Maize, Guanajuato, and Chiapas, Cycles 2012 and 2013, and Tlaxcala 2013** | | | **Group 2: Maize, Sonora, Cycles 2017–2020** | | | **Group 3: Wheat, Sonora, Cycles 2018–2019 and 2019–2020** | | | **Group 3: Wheat, Sonora, Cycles 2018–2019 and 2019–2020 (Varieties Pooled Together)** | | |
| *Source* | *df* | *p* | *Source* | *df* | *p* | *Source* | *df* | *p* | *Source* | *df* | *p* |
| State (S) | 2 | *** | Year (Y) | 3 | *** | Year (Y) | 1 | *** | Year (Y) | 1 | *** |
| Year (Y) | 1 | *** | Rep | 3 | ns | Rep | 3 | ** | Rep | 3 | * |
| Rep | 2 | ns | Treatment (T) | 7 | *** | Treatment (T) | 8 | ns | Treatment (T) | 8 | ns |
| Treatment (T) | 16 | *** | Y × T | 21 | ns | Var (V) | 1 | * | Y × T | 8 | ns |
| S × Y | 1 | *** | | | | Y × T | 8 | ns | | | |
| S × T | 32 | ** | | | | Y × V | 1 | *** | | | |
| Y × T | 16 | ns | | | | T × V | 8 | ns | | | |
| S × Y × T | 16 | ns | | | | Y × T × V | 8 | ns | | | |

| Non-grouped experiments. | | | | | | | | | | | |
|---|---|---|---|---|---|---|---|---|---|---|---|
| **Tlaxcala: Maize, cycle 2012** | | | **Campeche: Maize, cycle 2018** | | | **Campeche: Maize, cycle 2019** | | | | | |
| *Source* | *df* | *p* | *Source* | *df* | *p* | *Source* | *df* | *p* | | | |
| Rep | 2 | ns | Rep | 2 | ns | Rep | 2 | ns | | | |
| Treatment (T) | 12 | ns | Treatment (T) | 11 | ns | Treatment (T) | 9 | ns | | | |

\*, \*\*, \*\*\*, ns = Significant at 0.05, 0.01, <0.001, and non-significant, respectively.

Group 1 neither had a non-significant state × year × treatment interaction nor was it significant in the year × treatment interaction. However, the state × treatment interaction was significant. This interaction occurred due to a different magnitude of response of treatments across states. In Chiapas, a group of biofertilizers significantly out yielded the control treatment and yielded similarly to the full RFR. In contrast, no significant differences were observed for the control and the biofertilizer containing treatments in Guanajuato or Tlaxcala 2013 (Figure 1 and Table 8). In Tlaxcala 2013, the site was characterized by

drought during the reproductive stage (beginning of September). Furthermore, the date of seeding (18 of May) was later than what is locally recommended, due to a delay in the rainy season. The group of biofertilizers that resulted in higher yields in Chiapas, were BiofertiBUAP, MicorrizaFer, and FerbiliQ, which yielded an average of 1300 kg ha$^{-1}$ higher than the control (a 27% difference) and 790 kg ha$^{-1}$ less than the full RFR (a 12% difference). Thus, this group of biofertilizers in Chiapas fully compensated the reduction of 50% of synthetic fertilizers, as compared with the full local RFR. The yield difference between the full RFR and the control was 2090 kg ha$^{-1}$ higher for the full RFR than the control (a 43% difference). This substantial difference indicates the need of the RFR to achieve the maximum yields and highlight the merit of biofertilizers on achieving the maximum yields. The yield difference between the full RFR and non-fertilized treatment was 5099 kg ha$^{-1}$ higher for the full RFR than the non-fertilized treatment (a 383% difference). This large difference points out that the fertility level on the non-fertilized treatment was very low (high probability of yield response to fertilizers). In Guanajuato, no significant difference existed between the biofertilizers and control. However, a significant difference was observed between the full RFR and control, accounting for 1335 kg ha$^{-1}$ in favor of the full RFR (an 11% difference). Also significant was the comparison between the full RFR and non-fertilized treatment, recording a difference of 2568 kg ha$^{-1}$ (a 22% difference) (Figure 1 and Table 8). Tlaxcala 2013, probably caused by a very low soil organic matter (O. M.) content, with only 0.6% (Table 2), averaged a yield of 3207 kg ha$^{-1}$ and did not show significant differences, not only among biofertilizers and the control, but among any of the treatments (Figure 1 and Table 8).

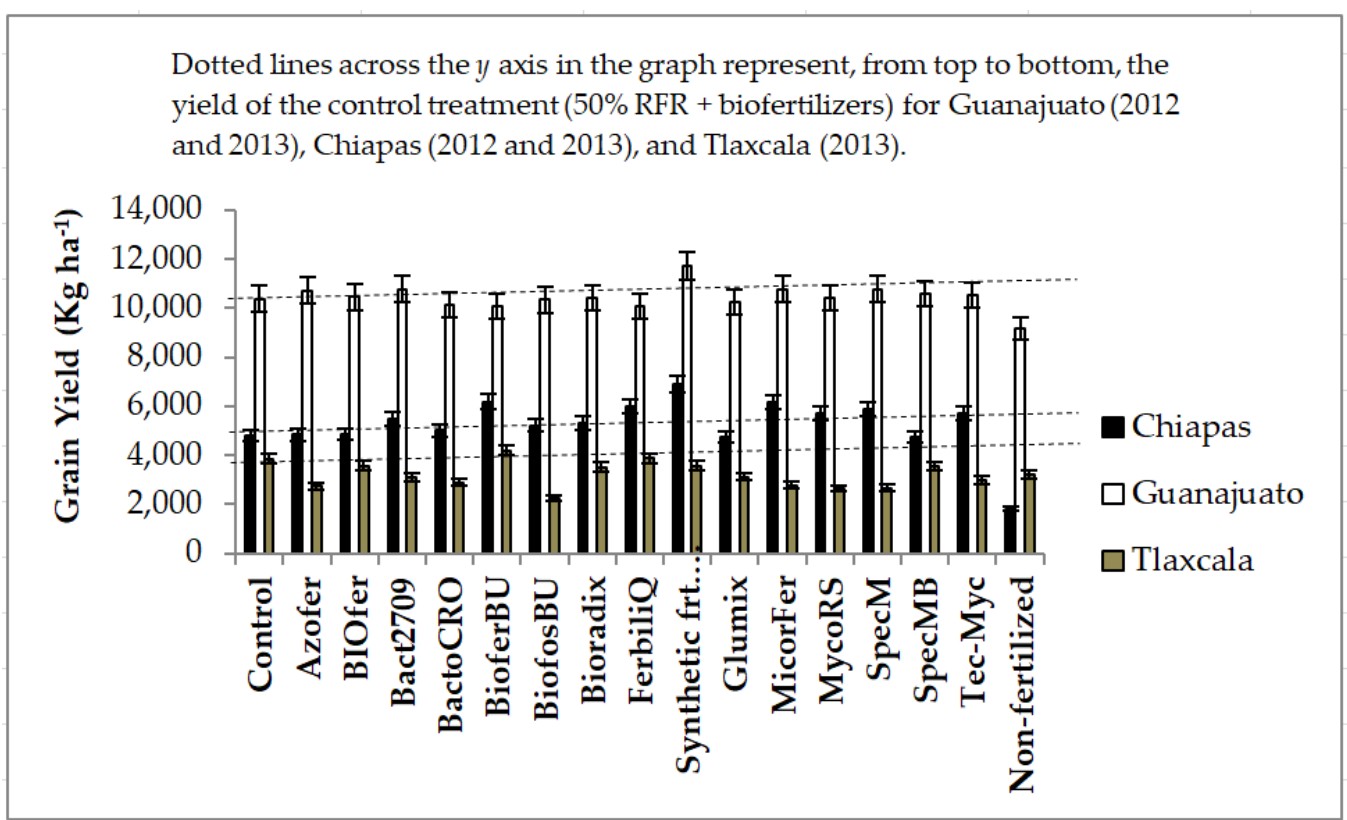

**Figure 1.** States × treatments interaction in three Mexican states (Group 1) to test the effectiveness of partially substituting synthetic fertilizers by biofertilizers.

**Table 8.** Means comparisons of experiments conducted in three Mexican states during 2012 and 2013 (Group 1) to test the effectiveness of partially substituting synthetic fertilizers by biofertilizers.

| Guanajuato (Maize-Cycles 2012 and 2013) | | | | | Chiapas (Maize-Cycles 2012 and 2013) | | | | | | | Tlaxcala (Maize-Cycle 2013) | | | |
|---|---|---|---|---|---|---|---|---|---|---|---|---|---|---|---|
| Treatments | *n* | Means (kg ha$^{-1}$) † | | | Treatments | *n* | Means (kg ha$^{-1}$) † | | | | | Treatments | *n* | Means (kg ha$^{-1}$) † | |
| Synthetic frt. 100% | 6 | 11,716 | | a | Synthetic frt. 100% | 6 | 6903 | a | | | | Azofer | 3 | 2740 | a |
| Spectrum Mico | 6 | 10,783 | b | a | BiofertiBUAP | 6 | 6180 | b | a | | | BIOfertilizante | 3 | 3575 | a |
| MicorrizaFer | 6 | 10,776 | b | a | MicorrizaFer | 6 | 6158 | b | a | | | Bacteriano 2709 | 3 | 3102 | a |
| Bacteriano 2709 | 6 | 10,765 | b | | FerbiliQ | 6 | 6001 | b | a | c | | BactoCROP-TS | 3 | 2890 | a |
| Azofer | 6 | 10,725 | b | | Spectrum Mico | 6 | 5880 | b d | a | c | | BiofertiBUAP | 3 | 4188 | a |
| Spectrum Mico Bac | 6 | 10,586 | b | | Mycor Root Saver | 6 | 5715 | b d | e | c | | BiofosfoBUAP | 3 | 2257 | a |
| Tec-Myc 60 | 6 | 10,537 | b | | Tec-Myc 60 | 6 | 5685 | b d | e | c | | Bioradix | 2 | 3521 | a |
| BIOfertilizante | 6 | 10,447 | b | | Bacteriano 2709 | 6 | 5479 | b d | e | c | | Non-fertilized | 3 | 3226 | a |
| Bioradix | 6 | 10,417 | b | | Bioradix | 6 | 5301 | b d | e | c | | FerbiliQ | 1 | 3869 | a |
| Mycor Root Saver | 6 | 10,414 | b | | BiofosfoBUAP | 6 | 5215 | b d | e | c | | Synthetic frt. 100% | 2 | 3573 | a |
| Synthetic frt. 50% | 6 | 10,381 | b | | BactoCROP-TS | 6 | 5009 | d | e | c | | Synthetic frt. 50% | 3 | 3843 | a |
| BiofosfoBUAP | 6 | 10,335 | b | | BIOfertilizante | 6 | 4853 | d | e | | | Glumix | 3 | 3116 | a |
| Glumix | 6 | 10,261 | b | | Azofer | 6 | 4838 | d | e | | | MicorrizaFer | 2 | 2771 | a |
| BactoCROP-TS | 6 | 10,126 | b | | Synthetic frt. 50% | 6 | 4813 | d | e | | | Mycor Root Saver | 2 | 2638 | a |
| FerbiliQ | 6 | 10,084 | b c | | Spectrum Mico Bac | 6 | 4739 | e | | | | Spectrum Mico | 3 | 2664 | a |
| BiofertiBUAP | 6 | 10,057 | b c | | Glumix | 6 | 4733 | e | | | | Spectrum Mico Bac | 3 | 3557 | a |
| Non-fertilized | 6 | 9148 | c | | Non-fertilized | 6 | 1805 | f | | | | Tec-Myc 60 | 1 | 2986 | a |
| *Mean* | | *10,445* | | | *Mean* | | *5253* | | | | | *Mean* | | *3207* | |
| *LSD* | | *950* | | | *LSD* | | *1137* | | | | | *LSD* | | *1995* | |
| *CV* | | *8* | | | *CV* | | *19* | | | | | *CV* | | *29* | |

† Means followed by different letters within a column are significantly different (*p* < 0.05). *n* = Number of observations in the means.

Group 2, which includes the four maize experiments in Sonora, showed a non-significant year × treatment interaction but a significant difference among treatments (Table 7). Table 9 shows the mean comparisons for Groups 2 and 3. In Group 2, the core comparison in this study, the biofertilizers vs. the control was significant in one case. The biofertilizer containing treatment of Azofer + MicorrizaFer, significantly out yielded the control. The difference between Azofer + MicorrizaFer and the control was 823 kg ha$^{-1}$ higher for the biofertilizers (a 9% difference). No other significant difference between the biofertilizers and control was recorded. The difference between the full RFR and control was 2110 kg ha$^{-1}$ higher for the full RFR (a 24% difference). This latter comparison shows that although there was ample margin for increasing yields (2100 kg ha$^{-1}$) by applying the full rate as compared with the control, only one biofertilizer containing treatment contributed to the yield recorded for the control treatment. Other than this exception, the biofertilizers did not contribute any additional yield, compared to the control. The difference between the full RFR versus the non-fertilized treatment was 7080 kg ha$^{-1}$ higher for the full RFR treatment (a 280% difference); this large difference indicates that the natural fertility of the experimental area was in great need of nitrogen fertilizers to support maximum yields.

**Table 9.** Means comparisons of experiments conducted in Sonora, Mexico from 2017–2020 with maize (Group 2) and from 2018–2019 and 2019–2020 with wheat (Group 3) to test the effectiveness of partially substituting synthetic fertilizers by biofertilizers.

| Group 2: Maize, Sonora, Cycles 2017–2020 | | | | | | Group 3: Wheat, Sonora, Cycles 2018–2019 and 2019–2020 (Varieties Pooled Together) | | | |
|---|---|---|---|---|---|---|---|---|---|
| Treatments | *n* | Means (kg ha$^{-1}$) † | | | | Treatments | *n* | Means (kg ha$^{-1}$) † | |
| Synthetic fertilizer 100% | 6 | 11,007 | a | | | Synthetic fertilizer 50% | 7 | 7291 | a |
| Azofer + MicorrizaFer | 6 | 9721 | | b | | Synthetic fertilizer 100% | 7 | 7248 | a |
| BiofosfoBUAP | 6 | 9123 | | b | c | BiofosfoBUAP | 7 | 7216 | a |
| BiofertiBUAP | 6 | 8914 | | | c | Non-fertilized treatment | 7 | 7177 | a |

**Table 9.** *Cont.*

| Group 2: Maize, Sonora, Cycles 2017–2020 | | | | Group 3: Wheat, Sonora, Cycles 2018–2019 and 2019–2020 (Varieties Pooled Together) | | | |
|---|---|---|---|---|---|---|---|
| Treatments | *n* | Means (kg ha$^{-1}$) † | | Treatments | *n* | Means (kg ha$^{-1}$) † | |
| Synthetic fertilizer 50% | 6 | 8898 | c | BiofertiBUAP | 7 | 7110 | a |
| Mycor Root Saver | 6 | 8829 | c | Azofer + MicorrizaFer | 7 | 7022 | a |
| FerbiliQ | 6 | 8799 | c | FerbiliQ | 7 | 6956 | a |
| Non-fertilized treatment | 6 | 3928 | d | Mycor Root Saver | 7 | 6927 | a |
| *Mean* | | 8653 | | BiofosfoBUAP + BiofertiBUAP | 7 | 6802 | a |
| *LSD* | | 707 | | *Mean* | | 7083 | |
| *CV* | | 12 | | *LSD* | | 500 | |
| | | | | *CV* | | 7 | |

† Means followed by different letters within a column are significantly different (*p* < 0.05). *n* = Number of observations in the means.

Group 3. Looking at the wheat experiments in Sonora we found a non-significant years × treatments × wheat varieties interaction, neither was the treatments × wheat varieties interaction significant, nor was the effect of treatments significant. Since no significant treatments × wheat varieties interaction was recorded, the factor of varieties was pooled together in a separate analysis (Table 7). Once the effect of varieties was pooled together, the effect of treatments still remained non-significant (Table 9). The means of Group 3 are shown in Table 9. The difference between the control and mean of all biofertilizers was 261 kg ha$^{-1}$ higher for the control (a 4% difference). The difference between the full RFR treatment and control was 44 kg ha$^{-1}$ higher for the control (less than 1% difference); and the difference between the full RFR treatment and non-fertilized treatment was 71 kg ka$^{-1}$ higher for the full RFR treatment (a 1% difference). The latter comparison shows that the residual fertility of the experimental area was high enough to maintain the maximum yields, providing little room for biofertilizers to demonstrate any advantages.

The experiments that were analyzed independently (non-grouped) were Tlaxcala 2012, Campeche 2018, and Campeche 2019. No significant treatment effect was recorded for any of these three experiments (Table 5). In Table 10, the means of all three experiments are shown. A possible explanation for a lack of significance among treatments in Tlaxcala 2012 is explained by the occurrence of consecutive dry years, being these rainfed experiments, together with the presence of the highest coefficient of variation, indicating high soil variability in the experimental area. Under such abnormal conditions in this state, the effect of using biofertilizers could not be determined.

**Table 10.** Means of non-grouped experiments in Tlaxcala in 2012 and in Campeche in 2018 and 2019, to test the effectiveness of partially substituting synthetic fertilizers by biofertilizers in Mexico.

| Tlaxcala: Maize, Cycle 2012 | | | | Campeche: Maize, Cycle 2018 | | | | Campeche: Maize, Cycle 2019 | | | |
|---|---|---|---|---|---|---|---|---|---|---|---|
| Treatments | *n* | Means (kg ha$^{-1}$) † | | Treatments | *n* | Means (kg ha$^{-1}$) † | | Treatments | *n* | Means (kg ha$^{-1}$) † | |
| Synthetic fertilizer 50% | 3 | 3335 | a | Synthetic frt. YARA (21-17-3-4) | 3 | 5751 | a | Bioradix | 3 | 6495 | a |
| MicorrizaFer | 3 | 2916 | a | Synthetic 120-55-48 | 3 | 4753 | a | Synthetic 120-55-48 | 3 | 6484 | a |
| BIOfertilizante | 3 | 2831 | a | Synthetic 90-66-48 | 3 | 4678 | a | Mycor Root Saver | 3 | 6328 | a |
| Synthetic frt. 100% | 3 | 2619 | a | Mycor Root Saver | 3 | 4660 | a | Synthetic 90-66-48 | 3 | 6101 | a |
| Non-fertilized | 3 | 2456 | a | Synthetic 27-69-00 | 3 | 4503 | a | MicorrizaFer + Azofer | 3 | 6089 | a |
| Bacteriano 2709 | 3 | 2380 | a | Bioradix | 3 | 4443 | a | Microbiología MICI | 3 | 6007 | a |
| FerbiliQ | 3 | 2205 | a | MicorrizaFer | 3 | 4399 | a | BiofosfoBUAP + BiofertiBUAP | 3 | 5577 | a |
| BiofosfoBUAP | 3 | 2194 | a | Tec-Myc 60 | 3 | 4380 | a | Synthetic 27-69-00 | 3 | 5436 | a |
| Mycor Root Saver | 3 | 2150 | a | BiofosfoBUAP | 3 | 4267 | a | YARA | 3 | 5420 | a |

**Table 10.** *Cont.*

| Tlaxcala: Maize, Cycle 2012 | | | | Campeche: Maize, Cycle 2018 | | | | Campeche: Maize, Cycle 2019 | | | |
|---|---|---|---|---|---|---|---|---|---|---|---|
| Treatments | *n* | Means (kg ha$^{-1}$) † | | Treatments | *n* | Means (kg ha$^{-1}$) † | | Treatments | *n* | Means (kg ha$^{-1}$) † | |
| Tec-Myc 60 | 3 | 1963 | a | FerbiliQ | 3 | 4249 | a | Synthetic 27-69-00 + PEPTON | 3 | 4730 | a |
| Azofer | 3 | 1786 | a | ISQUISA 13-08-16 | 3 | 4205 | a | *Mean* | | *5867* | |
| BactoCROP-TS | 3 | 1536 | a | Microbiología MICI | 3 | 4093 | a | *LSD* | | *1498* | |
| Bioradix | 3 | 1511 | a | *Mean* | | *4532* | | *CV* | | *15* | |
| *Mean* | | *2299* | | *LSD* | | *1240* | | | | | |
| *LSD* | | *1470* | | *CV* | | *16* | | | | | |
| *CV* | | *38* | | | | | | | | | |

† Means followed by different letters within a column are significantly different ($p < 0.05$). *n* = Number of observations in the means.

In Campeche, the effect of treatments was not significant in any of the two cycles, 2018 or 2019 (Table 5). In addition, in 2018, the contrast of the synthetic fertilizer treatment (90-66-48, N-P-K) against the average of all biofertilizer containing treatments plus 90-66-48 (N-P-K) had a $p = 0.4835$ and the same contrast in 2019 had a $p = 0.9972$. The means for Campeche are shown in Table 10. These results indicate that in this location, the use of biofertilizers did not result in higher yields, compared with the purely synthetic fertilizer treatments.

In summary, Table 11 shows the number of experiments where the biofertilizer had a significant yield response across locations, cycles, and crops.

**Table 11.** Number of significant biofertilizers yield responses by number of experiments tested across locations, cycles, and crops.

| Biofertilizer | Organisms in Biofertilizers | Significant Yield Response Number by Number of Tested Locations/Years † | Location of Positive Yield Response |
|---|---|---|---|
| Mycor Root Saver | *Entrophospora columbiana*, *Glomus* spp. | 0 of 8 | - |
| Azofer | *Azospirilum brasilense* | 1 of 7 | Sonora |
| MicorrizaFer | *Glomus* spp. | 2 of 8 | Chiapas, Sonora |
| FerbiliQ | *Azospirilum brasilense*, *Glomus intradices* | 1 of 7 | Chiapas |
| Tec-Myc 60 | *Glomus* spp., *Acaulospora scorbiculata*, *Gigaspora margarita*, *Bacillus subtillis*, *Azosprilum brasilense* | 0 of 5 | - |
| BactoCROP-TS | *Azospirillum* spp., *Bacillus* spp. | 0 of 4 | - |
| BiofertiBUAP | *Azospirillum* spp. | 1 of 7 | Chiapas |
| Bioradix | *Azospirilum brasilense* | 0 of 6 | - |
| Bacteriano 2709 | *Pseudomonas* spp. | 0 of 4 | - |
| BIOfertilizante | *Glomus intradices* | 0 of 4 | - |
| BiofosfoBUAP | *Pseudomonas* spp. | 0 of 9 | - |
| Spectrum Mico | *Glomus* spp. | 1 of 3 | Chiapas |
| Spectrum Mico Bac | *Glomus* spp., *Bacillus* spp. | 0 of 3 | - |
| Glumix | *Glomus geosporum*, *Glomus fasciculatum*, *Glomus constrictum*, *Glomus tortuosum*, *Glomus intradices* | 0 of 3 | - |
| Microbiologia MICI | Unspecified | 0 of 2 | - |

† The yield response in Sonora was a combination of Azosfer + MicorrizaFer and MicorrizaFer in Chiapas. Furthermore, the response was consistent throughout the years (i.e., consistent results).

## 4. Discussion

It was found that there are significant benefits of the biofertilizer MicorrizaFer in Sonora and Chiapas (AMF); locations with low P soils with maize. Given the extent of low P tropical soils in maize production systems in Mexico, this is a very relevant finding. Other than that, in general, no response to biofertilizers was observed in the rest of the locations through the years in maize. The yield response in Chiapas in this study coincides with that reported also in Chiapas by Martínez-Reyes et al. [3], who found a positive response of maize yield to a combination of synthetic fertilizer plus biofertilizers. It is worthwhile to notice that both the Chiapas and Tlaxcala sites had moderately low soil P availability. The benefits of AMF symbiosis in maize in low P soils have been well documented [45,46]. In contrast, the different response of wheat to AMF under low-moderate P has also been documented. Since wheat has a more extensive root system and root exudates, it is less dependent on AMF for P uptake and the response is generally less; while, maize is a crop that is highly dependent on AMF for P uptake due to root architecture [47]. A generalized lack of wheat yield response to biofertilizers in the present study coincides with another series of experiments conducted in Guanajuato, Mexico by Rodríguez-Ramos et al. [43] with an inconsistent response of biofertilizers in wheat. Across all experiments, this study concluded that the addition of biofertilizers had an inconsistent response on yield and that, in general, did not compensate cereal yields due to reductions on the recommended synthetic fertilizer rates.

In experiments conducted under irrigation in Guanajuato and Sonora, excluding the one with maize in Sonora, which positively responded to biofertilizers (experiment that demonstrated being nitrogen deficient), it is suggested that the lack of yield response to biofertilizers could be partly be explained by the relatively high soil fertility. In Tlaxcala, there was no difference between the 50% and 100% RFR. There was no additional yield response from the 100% RFR or biofertilizers; indicating plant nutrition limitations were covered by the 50% rate under water limiting rainfed conditions. In Guanajuato, on the other hand, the unfertilized yield was over 9 t ha$^{-1}$; indicating high residual soil fertility. High soil fertility could be the result of the residual effect of nutrients caused by the continuous application of considerable high fertilizer rates through the years that are typically employed in these intensive production systems. Sonora ranks first in wheat production in Mexico and Bajío (Guanajuato) ranks second or third every year. Thus, these states are highly productive and use high level of inputs for crop production. Even though in the present study the full recommended fertilizer rates were cut by half, soil fertility still remained relatively high, as observed by the small or almost null gap between the full RFR and the non-fertilized treatments recorded for maize in Guanajuato and wheat in Sonora (Tables 8 and 9). Thus, in these two locations, where relatively high fertility may occur, biofertilizers may not have been as effective in high fertility environments as in low fertility environments like in Chiapas or Sonora, in the case of maize, where the gap between the full RFR and the non-fertilized treatment was a 383% and 280% difference, respectively.

In a classic paper, Fuentes-Ramirez and Caballero-Mellado [42], reported results from an extensive campaign where biofertilizers were tested in Mexico. They reported that "When nitrogen fertilizers were not applied to traditional and modern maize cultivars, the inoculation with Azospirillum exerted beneficial effects in 95 and 93% of the sites evaluated during 1999 and 2000, respectively. However, when fertilizers were applied in levels higher than 110 kg N/ha, the positive responses on the maize yield were observed only in 55 and 50% of the sites evaluated in 1999 and 2000, respectively". Banayo et al. [48] supported the hypothesis of the inconsistency of biofertilizers performance in the Philippines due to relatively high fertility levels in rice production system. They concluded that " . . . the trends in our results seem to indicate that biofertilizers might be most helpful in rainfed environments with limited inorganic fertilizer input". At the 50% RFR, the nitrogen fertilizer rates in Guanajuato and Sonora were 100 and 125 kg N ha$^{-1}$, respectively, on top of the modest residual fertility levels (Table 2) (modest residual fertility if 35 kg N are required to produce 1000 kg of wheat, in environments where mean yields are around 6500 kg ha$^{-1}$).

Even these slightly high fertility levels could have inhibited the response of microorganisms contained in the applied biofertilizers. In further support of this hypothesis about high soil fertility in Guanajuato and Sonora cancelling the benefits of biofertilizers to crop yields, Fukami et al. [23] observed that the efficiency of *Azospirillum* spp. to support crop yields depended on N rates. High N rates would decrease the ability of *Azospirillum* spp. to promote positive responses on crop yields, due to a decreased activity of the enzyme nitrogenase, while at low N rates its ability to stimulate a positive response is increased. These results are in general agreement with those reported by Ramírez-Ramos et al. [43], where, with the exception of the results observed in Villagrán, there was no effect of the biofertilizers on wheat or barley (*Hordeum vulgare*) yields in Guanajuato. The present study supports the hypothesis of high soil fertility as yields of maize without fertilizers were high, as compared to the 100% RFR (Table 8). In addition, relatively high soil supplies of available phosphorus (P) may have, as well as N, inhibited the response to mycorrhizas-based biofertilizers. Davaran et al. [20] found a negative interaction between *Glomus* spp. and P fertilization at levels higher than 50% the locally recommended rate (equivalent to 50 kg $P_2O_5$ ha$^{-1}$), while the mean $P_2O_5$ applied fertilizer rate in non-responsive sites in the present study was 54 kg $P_2O_5$ ha$^{-1}$ on top of the P in soil residual reserves (Table 2). Jensen and Jacobson [47] provided additional evidence about vesicular-arbuscular mycorrhizas being inhibited by high P levels in soil and *vice versa*.

On the other hand, a lack of an adequate water supply in rainfed experiments, except for Chiapas where annual precipitation exceeds 1000 mm, could have been the main reason for the lack of response of maize to biofertilizers. Glazova [48] reported that the efficiency of bio-fertilization directly depended on soil moisture levels, being the optimum at a level as high as 60% soil moisture. In addition, Alahdadi et al. [49] reported a significant water deficit stress × cultivar × biofertilizer interaction on soybeans (*Glycine max* L.). They suggested that by increasing the severity of water deficit stress, the primary root length decreased. This could be the result of a disruption in photosynthesis because of the shortage of soil moisture and decreasing transport of photosynthates to the plant during the growth period.

Cassán et al. [25] point to a number of possible reasons for restricting the response of biofertilizers in different crops. Crop plant root factors such as surface area, root hair abundance and length, growth rate, response to soil conditions, and exudations determine the relative dependency on AMF for nutrient uptake [45,46]. Other causes include complex interactions between microorganisms in biofertilizers and plants; strong stressful crop growing conditions; unsound methods of inoculation; a lack of replicability of experimental conditions; and the interaction of native soil biota with inoculants, i.e., studies under isolated controlled conditions often produce different results under field uncontrolled conditions. Other suggested reasons for the lack of response to biofertilizers include possibly a poor quality of biofertilizer standards [29]. In the present study, for example, Azofer + Microrriza Fer (*Glomus* spp. based) out yielded the control in Sonora and Chiapas; both maize and low available P sites, but the same biofertilizer, sold by the same company, did not show a yield response in any other part of the environment, which may suggest variability in quality of products from different manufacturers, although there are other several factors, inherent to individual biofertilizer users that may damage the product such as prolonged exposure time of biofertilizer products or biofertilizer treated seed to direct sunlight in the field (among a number of other particular practices and environmental conditions. Chávez-Díaz et al. [1] underlined that there are several factor interacting that need to be considered in order to take advantage of biofertilizers (Figure 2).

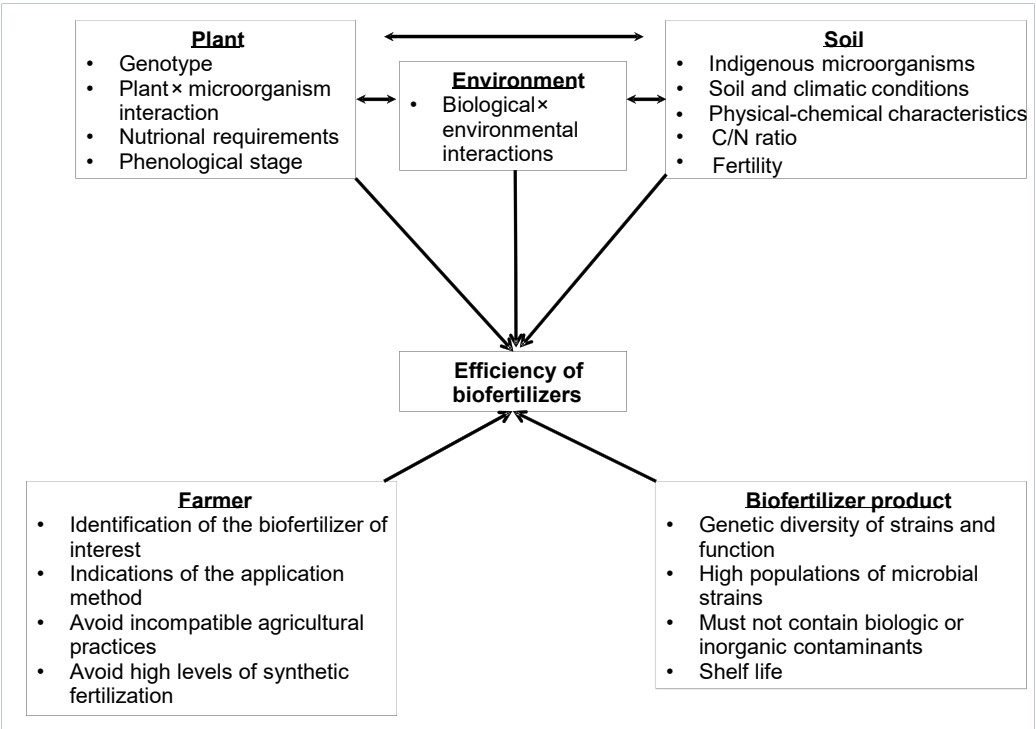

**Figure 2.** Considerations for successful use of biofertilizers in agriculture. Adapted from Chávez-Diaz et al. [1].

The use of biofertilizers is no doubt, less harmless to the environment than synthetic fertilizers, which is of key importance in today's world. However, the need of producing food for ≈ 8 billion people (as today, 14 December 2021) is equally or more important and requires increasing the productivity of crops, but crop productivity is unavoidably linked to substantial input use. It is a physics rule. The key for intensive production systems is to find ways to be more sustainable. This objective is possible without sacrificing crop yields. Minimum tillage systems, the use of tools to reasonable apply minerals, such the GreenSeeker® technology, which is capable of estimating the N needs of crops based on actual yield potential, among many other technologies are sound tools that are designed to make modern agriculture productive but friendlier to the environment, to farmer's income, and to society in general. The downside of biofertilizers is their lack of consistency across products, locations, and years. In addition, the interactions among these factors are complex to understand and apply at the field level, as biological processes are very dynamic in space and time for the microorganisms and for the plant's environments.

## 5. Conclusions

The results of these experiments show that only in Chiapas and Sonora (maize) was there was a significant increase in yield with some biofertilizers in combination with inorganic fertilizers. Therefore, one of the main conclusions of this study is that biofertilizers only work in some places and in the places where biofertilizers show a response, only some of them work. These results suggest that we should be cautious before widely recommending the use of biofertilizers across Mexico since their positive response on yields seem to be more of an exception rather than the rule.

From the results observed in the present multiple-locations (with highly contrasting agroecologic characteristics and input use levels), multiple-year study, it is suggested that the lack of response to some environments may be related to the level of precipitation, organic matter content, and residual soil fertility. Further research is needed to test biofertilizers in environments representative of smaller farmers, with even lower input investment and surely more responsiveness to fertilizer applications, than those represented in the

current research, since, Tlaxcala and Chiapas experiments were developed under medium input, while Sonora and Guanajuato were high input production systems. Another lesson the experiments left us is, in future research, to include another control treatment with, perhaps, 25% of the locally RFR, since the 50% RFR often yielded the same as the full RFR, indicating that in these locations we did not lower the fertility level to achieve a more accurate evaluation of the efficacy of biofertilizers.

While we found evidence that there can be benefit from some products in low-moderate soil fertility conditions, there were by far more products that provided no benefit and resulted in yield loss. Only four of the 15 biofertilizer products produced a yield response and only one in more than one location (MicorrizaFer). While the benefits of a biofertilizer can be significant under a certain condition with the right product, there is a greater chance of a farmer using a product with no benefit. This shows the need for well-designed field trails in experimental platforms to test products before recommending to farmers to avoid risk of yield loss. Farmers need clear guidance on the use of biofertilizers, what products are recommended, and how much synthetic fertilizer rates can be reduced. Further research on this is required to fine tune recommendations to maximize yields and economic benefits for farmers.

**Author Contributions:** Conceptualization, M.-S.T. and I.O.-M.; Data curation, J.S.-C., M.-S.T., M.E.C.-C., L.G.-Z. and I.O.-M.; Funding acquisition, I.O.-M.; Investigation, M.-S.T.; M.E.C.-C., S.M.-P., A.L.-O., R.P.-M., L.G.-Z. and I.O.-M.; Methodology, M.-S.T. and I.O.-M.; Project administration, M.E.C.-C.; Resources, I.O.-M.; Visualization, J.S.-C.; Writing-original draft, J.S.-C.; Writing-review & editing, J.S.-C., M.-S.T., A.L.-O., R.P.-M. and I.O.-M. All authors have read and agreed to the published version of the manuscript.

**Funding:** This work was implemented by CIMMYT as part of the project "Cultivos paraMéxico/MasAgro", made possible by the generous support of SADER, Mexico. We would like to acknowledge the financial support from CCAFS (Climate Change Agriculture and Food Security).

**Institutional Review Board Statement:** The present study did not require ethical approval.

**Informed Consent Statement:** Not applicable.

**Data Availability Statement:** All data reported in the present study can be shared upon request by Iván Ortiz-Monasterio at: i.ortiz-monasterio@cgiar.org.

**Acknowledgments:** The authors acknowledge the support from Delia Gallegos-Apodaca, Research assistant at CIMMYT, for the elaboration of Tables about climatic conditions in the different sites of research. The authors also express their appreciation to Carlos Augusto Tapia-Moo, for conducting the experiments in Campeche.

**Conflicts of Interest:** Although the biofertilizers tested in the present research were donated by private companies, the authors declare no conflict of interest, since the donor companies had no role in the design of the study; in the collection, analyses, or interpretation of data; in the writing of the manuscript, or in the decision to publish the results.

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
