# Peer review of "Can Biofertilizers Reduce Synthetic Fertilizer Application Rates in Cereal Production in Mexico?"

_agronomy, doi:10.3390/agronomy12010080_

Round 1
Reviewer 1 Report
Dear Authors,
The work is very extensive and incomprehensible. It is hard to read. I have a number of doubts and comments. The authors in Table 2 report the results in ppm, meq/100g or%. In the MDPI manuscripts, the SI units mg kg-1 or g kg-1 are preferred. The abbreviations should be given in the table description (CIC, nil?) Why are the contents given in meq / 100g for Sonoran locations? This causes confusion! Campeche - n.a. why is this object included in the presented research? Why did the authors not provide the weather conditions as mean values ​​for the study periods for the locations? This information would be clearer. In table 3, seeding density is listed once as plants ha-1, or plants m-1 or 120 kg ha-1 for Cirno C-2008 and 100 kg ha-1 for Borlaug-100 - what does it mean? How can fertilization objects be compared where the RFR is different for the same plant? For example N fertilization from 90 to 240 kg ha-1? Fertilization: urea as nitrogen source, diammonium phosphate as a phosphorus source, and potassium chloride as potassium source. Isn't diammonium phosphate also a source of nitrogen? Why are tables copied from Excel? They should be carefully and clearly prepared. How can the authors compare the effect of bio-fertilizers if they use them on different cultivars with different doses of fertilizers? The methodology is described in an incomprehensible way. Too much data, work should be limited to comparable objects! In the Discussion section, there are many citations from other works. Here authors should find comparisons between their own research and the results of other authors. In this section, we do not describe the studies described in other papers listed in the References section. This manuscript is intended to be an Article, not a Review Paper. Why are all tables and figures included in Appendix B? Are they not the same as those given in the text of the manuscript?
Kind Regards
Reviewer
Author Response
Dear Reviewer;
Thank you for the time and effort to review our Manuscript.
We have addressed your observation carefully, one by one and are sure the new version has improved as a result of this revision. Thank you, again.
Please see the attachment.

Reviewer 2 Report
Please, check the file attached.

Author Response

(The authors gave the same response as above.)

Reviewer 3 Report
The paper reports results on the effectiveness of biofertilizers compared to standard chemical fertilizer. The results are from numerous states in Mexico, from multiple sites, over a number of years and tested multiple biofertilisers, so are very comprehensive. Overall, the results are interesting because they give a reliable indication of how effective biofertilizers can be in a range of climatic and soil conditions. The main conclusion is that the effectiveness is variable, and depends mostly on the existing soil fertility, and the type of biofertilizer applied, so it is a complicated process to achieve successfully. It was good that there were some significant positive results of the bioferts. I agree that a treatment with 25% chemical fertiliser might have been more informative, but the results are still useful and valid nonetheless. I think it was well written and clear, but I just have a few minor recommendations below.
Abstract and Introduction: OK
Methods:
Figs. 2 and 3 seem superfluous.
Table 4. next to organism column could have a column saying what each organism does for the crop e.g. AMF for P solubility, and which ones affect N-fixation.
Results:
Table 2. NA means not applicable not not available, so you have used NA incorrectly here
Figure 4. lots of issues:
- need a better order of treatments on x-axis to differentiate the control and non-chemical fert from biofert treatments.
- striped bars not good on the eyes
- y-axis title should include "yield"
- the horizontal control yield bars aren't very visible and have shifted along.
- need gaps between the bars of each treatment
- figure legend needs more detail.
Table 6: say that letters in final column indicate significant differences.
Tables 7 and 8: possibly also better as bar charts? Follow advice above for fig. 4.
Discussion:
Add reference to methods tables/figures in text.
Paragraph 1: why would AMF be affected by soil OC e.g. lines 321-325, AMF gets its carbon/energy from the plant, so this doesn't makes sense and needs better explanation. This whole paragraph needs more work and better explaining why the AMF fertilisers had affects at these sites.
Paragraphs 2 and 3/ lines 336-368, too much description about this experiment, need to summarise the main result.
Author Response

(The authors gave the same response as above.)

Round 2
Reviewer 1 Report
The authors corrected the manuscript which can now be considered printable. However, it is still difficult to read. The tables with the corrections made are hardly legible. I hope that the editors will deal with the appropriate formatting of the text